# Barriers to and enablers of childhood immunization uptake in Ethiopia's Amhara, Oromia, and Somali Regions: A multi-perspective qualitative study

Amare Zewdie[1], Minyahil Tadesse Boltena[1,2], Mengistu Ayenew[1], Tamrat Endebu[1], Melat Dereje[1], Alemseged Abdissa[1], Mirafe Solomon[3], Paula Valentine[4], Tahlil Ahmed[4], Andrew Clarke[4], Sostine Makunja[4], Mervat Alhaffar[5], Catherine R. McGowan[5], Nada Abdelmagid[5]*, Yohannes Hailemichael[1]

**1** Armauer Hansen Research Institute, Ministry of Health, Addis Ababa, Ethiopia, **2** Faculty of Public Health, Ethiopian Evidence-Based Health Care Centre: A Joanna Briggs Institute's Center of Excellence, Institute of Health, Jimma University, Jimma, Ethiopia, **3** Save the Children, Ethiopia Country Office, Addis Ababa, Ethiopia, **4** Save the Children UK, Global Programmes, London, United Kingdom, **5** Department of Infectious Disease Epidemiology and International Health, Faculty of Epidemiology and Population Health, London School of Hygiene and Tropical Medicine, London, United Kingdom

* nada.abdelmagid@lshtm.ac.uk

## Abstract

Ethiopia has the fourth-highest number of zero-dose children globally. Negative experiences and perceptions of immunization are recognized barriers to vaccination uptake but warrant context-specific investigation. We explored barriers and enablers to immunization uptake in selected regions of Ethiopia. We conducted a formative qualitative study involving 18 focus group discussions with men and women, in-depth interviews with 23 mothers of children with varying immunization statuses, and 42 key informant interviews with religious and community leaders and health workers in eight districts in Amhara, Oromia, and Somali regions. We identified shared and regionally distinct barriers. Common barriers included limited access to services in hard-to-reach areas, low awareness of immunization, competing household responsibilities for mothers, fear of side effects, and a lack of compassionate and respectful care from health workers. Forgetting vaccination appointments was frequently reported in Amhara and Oromia. In Amhara and Somali, mistrust of vaccinators and infrequent vaccination sessions were salient challenges. In Amhara, some believed that envy or praise by vaccinators could bring harm or misfortune to children, and that vaccination should be delayed until after baptism. In Oromia, beliefs that vaccines aggravate illness and that infants should not leave home before six months of age were reported. In Somali, perceived parental negligence and beliefs that vaccines are unnecessary were described. Engaging community, traditional, and religious leaders and fathers in immunization activities in Oromia, and aligning vaccination sessions with local holidays in Amhara, emerged as promising practices. Our findings show

**Data availability statement:** To access the data, requests should be submitted to the Armauer Hansen Research Institute Ethics Review Committee by email at the following email address: ahri.alerterc@ahri.gov.et.

**Funding:** This study was funded by a grant from GSK to the Save the Children Fund (https://www.gsk-savethechildren.com/) [author initials: MS, PV, TA, AC, SM], and a subgrant from Save the Children to LSHTM and AHRI (grant number 104369) [author initials: AZ, MTB, MAy, TE, MD, AA, MAI, CRM, NA, YH]. GSK had no role in study design, data collection and analysis, decision to publish, or preparation of the manuscript.

**Competing interests:** The authors have declared that no competing interests exist.

that knowledge and perceptions of vaccines, cultural norms, service accessibility, and experiences with vaccine-preventable diseases (VPDs) and vaccination can either encourage or discourage uptake. We recommend enhancing service delivery, improving caregiver interactions, and implementing two-way community engagement involving religious and community leaders, and caregivers of fully vaccinated children, with a focus on highlighting reductions in VPD burden.

## Introduction

In 2018, approximately 700,000 child deaths were attributed to vaccine-preventable diseases (VPDs), with about 99% of these fatalities occurring in low- and middle-income countries [1]. Furthermore, it is projected that around 52 million children under the age of five years will die from preventable causes by 2030, unless urgent measures, such as improving immunization coverage, are implemented [2]. Given the holistic impact of childhood immunization on the health, economic, and social well-being of populations, the World Health Organization and its member states are actively working to enhance and strengthen immunization coverage [3,4]. Despite these global and national efforts, reaching children in need of vaccination remains a significant challenge in low- and middle-income countries, including many in Africa [5].

In 2019, only 44% of children aged 12–23 months received all basic vaccines in Ethiopia, while approximately 19% remained entirely unvaccinated [6]. Ethiopia ranks fourth globally in the number of zero-dose children - those who have not received even a single dose of diphtheria-tetanus-pertussis (DTP)-containing vaccines - with 1.1 million unvaccinated children in 2023 [7]. This accounts for a significant portion of the 7 million zero-dose children in the African region and 14.3 million globally, as estimated in 2023 [7]. This substantial number of unvaccinated children increases the risk of disease outbreaks, preventable child deaths, and poor developmental outcomes [8–10]. Moreover, VPDs often result in significant financial losses, reducing parental productivity and straining limited healthcare resources [11].

Several national and global initiatives have sought to improve access to immunization services, including the United Nations' Sustainable Development Goals (SDGs) [12], the establishment of Gavi, the Vaccine Alliance [13], and the Immunization Agenda 2030 (IA2030) [3]. In Ethiopia, these commitments are reflected in the National Expanded Programme on Immunization Comprehensive Multi-Year Plan 2021–2025, which outlines strategies to increase coverage and reduce inequities [14]. Evidence of political commitment includes the implementation of the Reaching Every District (RED) approach, Sustainable Outreach Services (SOS), Periodic Intensification of Routine Immunization (PIRI), Child Health Day events, and pulse vaccination campaigns [14]. To address persistently low coverage among pastoralist populations, the Health Extension Programme (HEP) was adapted to improve access for mobile and hard-to-reach communities [15]. Following declines

in coverage during the COVID-19 pandemic, Ethiopia also developed national catch-up vaccination guidelines in 2021 and initiated nationwide implementation in 2022, further demonstrating renewed prioritization of the immunization programme [16].

Despite political commitment and government-led efforts to address underimmunization, both demand- and supply-side factors significantly influence the utilization of immunization services in Ethiopia. On the demand side, barriers include negative service perceptions and experiences, lack of awareness, misinformation, fear of side effects, caregivers' workloads, missed appointments, frequent displacement and low household decision-making power amongst mothers [17–19]. On the supply side, challenges include service unavailability, such as refusing to open vaccine vials for a few children, insufficient or absent health extension workers, vaccine shortages, service disruptions, inadequate tracking of defaulters, low community engagement, and poor documentation [20,21]. Quantitative studies have also indicated that factors such as maternal use of other health services (e.g., antenatal care and institutional delivery), knowledge about immunization, proximity to health facilities, maternal decision-making autonomy and higher income levels are positively associated with the completion of immunization among eligible children in Ethiopia [22–25].

Ethiopia is home to a diverse range of cultures, each with distinct practices that impact community health. The Expanded Programme on Immunization (EPI) was launched in 1980, initially providing six antigens as part of the routine immunization schedule and gradually expanding to cover 12 antigens today [14]. Immunization services are a core component of primary health care and are delivered through all public health facilities nationwide. At the grassroots level, health extension workers work closely with local administrative structures, particularly the Health Development Army (HDA), a large volunteer-based initiative inspired by military organization that mobilizes women to promote health behaviours such as vaccination, family planning, and hygiene within their neighbourhoods [26]. The HDA acts as a critical intermediary between formal health services and households, significantly enhancing the HEP and improving maternal and child health outcomes [16].

National analyses of immunization uptake reveal significant variation in supply and demand barriers across different regions. As a result, exploring these differences is essential to triangulate and strengthen the evidence for tailored interventions [27]. The three regions, Amhara, Oromia, and Somali, which together account for over two-thirds of the country's population, represent 80.2% of Ethiopia's zero-dose children [6], yet have diverse cultural, geographical, epidemiological, and health system contexts. Given other contextual factors, such as the prevalence of VPDs, internal displacement, and recurring conflict and drought in these regions, research is needed to investigate the specific challenges and opportunities for improving immunization service utilization. This formative study, therefore, aims to explore the barriers and enablers of immunization uptake from the perspective of different stakeholders in eight districts across these three regions in Ethiopia. The findings from this study informed the design of a large-scale intervention program to improve equitable immunization coverage in the selected study sites [28].

## Materials and methods

### Ethics statement

The study received approval by the Armauer Hansen Research Institute (AHRI) Institutional Ethics Review Board (number PO-08–24) and the London School of Hygiene and Tropical Medicine's Research Ethics Committee (number 29802). The study was conducted according to the principles expressed in the Declaration of Helsinki. Written informed consent was obtained from all participants.

### Study design

We conducted a formative qualitative study using focus group discussions (FGDs), in-depth interviews (IDIs), and key informant interviews (KIIs). This approach allowed us to identify the social, behavioural, cultural, and health service-related barriers and enablers from a diverse group of stakeholders in the study areas. Formative qualitative research

methodology is crucial for exploring stakeholders' experiences and insights in-depth, and for informing the design of targeted interventions grounded in the lived experiences of those involved [29].

## Study setting

We conducted the study during March and April 2024 in eight districts across three regions in Ethiopia. The regions and districts were purposively selected by the agency implementing the intervention and the Ethiopian Ministry of Health based on the number of zero-dose and under-immunized children, the occurrence of vaccine-preventable diseases such as measles, and the presence of displaced or migratory or conflict- or drought-affected populations. The study was conducted to explore the most appropriate design for an already planned large-scale intervention to improve equitable immunization coverage. The characteristics of the three regions and the names of the study districts (*woredas*) are presented in Table 1.

## Study participants

Study participants included caregivers (both men and women) of zero-dose, under-immunized, and fully immunized children. We included leaders of community-based organizations and representatives of people with disabilities and internally displaced people (IDPs). We also included a wide range of stakeholders, namely, political, administrative and religious leaders, community leaders, social influencers, traditional healers, and health workers. This diverse representation ensured that a broad spectrum of voices was heard, helping to identify specific concerns and insights related to barriers and enablers of immunization uptake.

A total of 18 FGDs were conducted across three different social groups: 8 FGDs with mothers (caregivers), 8 FGDs with men (including fathers) from the community, and 2 FGDs with young mothers aged 15–17 years. Each FGD comprised 6–10 participants.

A total of 23 IDIs were conducted with mothers of children at various immunization stages, including mothers of zero-dose children *(n = 6)*, under-immunized children *(n = 5)*, fully immunized children *(n = 6)*, and children who are currently up-to-date but have not yet completed their immunization schedule *(n = 6)*.

Forty-two key informant interviews (KIIs) were conducted with various stakeholders across different levels, including village "*kebele*" level community leaders *(n = 7)*, district-level officials *(n = 7)*, women leaders and youth influencers *(n = 8)*, religious leaders and traditional healers *(n = 8)*, staff from community-based organizations, and representatives of people with disabilities and IDPs *(n = 5)*, and health workers working on immunization *(n = 7)*.

In each selected district, three *kebeles* (the smallest administrative units in Ethiopia) were purposively chosen, comprising one urban and two rural *kebeles*. Participants for interviews were purposively selected by the research team, guided by health extension workers and community leaders in each kebele. The selection criteria for IDI and FGD participants

**Table 1. Characteristics of study regions.**

| Region | Geographic area (km²) | Number of administrative zones (sub-regions) | Number of districts (*woredas*) | Projected population (2022) | Number of hospitals | Number of health centres | Number of health posts | Districts (*woredas*) included in the study |
|---|---|---|---|---|---|---|---|---|
| **Amhara** | 170,752 | 14 | 139 | 22,877,365 | 88 | 872 | 3,565 | Wadila Dehana |
| **Oromia** | 353,690 | 21 | 286 | 39,980,837 | 109 | 1411 | 7,090 | Seweyna Guradamole |
| **Somali** | 350,000 | 11 | 97 | 6,506,235 | 13 | 215 | 1,327 | Shilabo Tuliguled Harshin Fik |

were their children's immunization status, as reported by their families and from the health facilities in the kebeles. For KIIs, we selected participants based on their roles in immunization service delivery, and their social influence within the community. The final sample size for each method and participant group was determined by data saturation.

## Data collection

Data were collected through FGDs, IDIs, and KIIs using semi-structured guides tailored for each method (S1 Text). These guides were adapted from a previous study that assessed the behavioural, socioeconomic, and health system determinants of immunization service utilization in Ethiopia [30] as well as the WHO's behavioural and social drivers tools for childhood vaccination [31]. The English versions of the guides were translated into Amharic, Afan Oromo and Somali languages by academics who are fluent in these languages.

The data were collected by sixteen experienced and trained qualitative research assistants, recruited from universities in the respective regions, who are fluent in local languages, and who hold Master's degrees. The research team consisted of three trainers (PhD, MSc) and seven data collectors (MPH, MSc) in Somali, two trainers (MPH) and five data collectors (MPH, MSc) in Oromia, and two trainers (MSc, MPH) and four data collectors (MPH, MSc) in Amhara. The data collectors were three females and thirteen males. The data collectors critically examined their own biases and experiences, and how these might influence the research process and findings, through self-reflection and discussion among themselves. The data collectors received three days of training in each region, covering data collection techniques, interview guides, and informed consent procedures.

Topics covered during the interviews and FGDs include barriers and facilitators to immunization service utilization, particularly for zero-dose children, traditional practices and beliefs related to immunization uptake, caregivers' knowledge and attitude towards vaccination, the role of community leaders and traditional healers in immunization, and perceptions and opinions of proposed intervention strategies. Participants' demographic information was also collected during the discussion or interview.

Interviews (IDIs, KIIs) were conducted at participants' homes, health centres, health posts, or workplaces to ensure they felt comfortable. The interview locations were mutually agreed upon by the interviewee and the interviewer. FGDs were held in health facilities and communal spaces such as village "*kebele*" compounds, health centres, and health posts. All interviews and discussions were voice-recorded. In addition, field notes were taken during the discussions.

Two researchers facilitated each FGD: one guided the discussion while the other took notes on non-verbal expressions and group interaction, which provided more detailed descriptions and interpretations of the subject matter. IDIs and KIIs typically lasted between 30 and 60 minutes, and the FGDs between 45 and 60 minutes. Participants were selected purposively to include individuals with diverse ages, genders, work roles, residences, and levels of social influence, thereby helping us identify both common and unique perspectives and experiences within our study population.

To ensure the trustworthiness – i.e., the overall quality, integrity, and believability - of the study, we aligned the data collection and analysis procedures with the four core principles of qualitative rigour: Credibility, Dependability, Confirmability, and Transferability [32]. To maintain the quality of evidence, the interview guides were pre-tested and adjusted accordingly. Minor adjustments were made to the wording of prompts and questions in each region. Data collectors received three days of training on both the study procedures and subject matter. The same experienced researchers were involved in both data collection and transcription, allowing them to recall the overall context—such as nonverbal expressions, emotions, and interactions—while capturing the participants' spoken words. Notes and memos were made during the interviews and FGDs by research team members to support the analysis. Moreover, an audit trail was maintained for all audio files and their corresponding transcriptions. A workshop was held with key stakeholders to share and validate the preliminary findings.

## Data management and analysis

All voice-recorded files, transcripts and summary notes were stored in REDcap (Version 13.10.1) electronic data capture tools hosted at AHRI [33,34]. Audio recordings were transcribed into English by experienced researchers who participated in the data collection process. After an initial reading of the full transcripts and a review of the interview guides, a codebook was developed by the research team members (AZ, MTB, M Ayenew, and YH). The codebook included the code, its definition, a representative text example from the transcripts, and the corresponding subtheme and main theme. A deductive and inductive coding approach was applied using MAXQDA (version 2020) by AZ and MTB [35]. The codebook served as a reference throughout the coding process and guided the presentation of the results.

The research team identified main themes and subthemes based on their interrelationships and analysed the data thematically. The findings were then reported, accompanied by detailed descriptions of these themes and their corresponding subthemes. We included direct participant quotes to contextualize the findings and provide readers with vivid insights. Study methods were reported following the Consolidated Criteria for Reporting Qualitative Research (COREQ) Reporting Qualitative Research (COREQ) [36] (S2 Text).

## Ethical considerations

In addition to ethical approvals, support letters were secured from Ethiopia's Federal Ministry of Health, Regional Health Bureaus in Amhara, Oromia, and Somali, as well as the Amhara Public Health Institute. Before participating, all study participants were provided with an information sheet that explained the study's purpose, objectives, risks, and benefits. Written informed consent was then obtained from each participant. Participants were also told that they could withdraw their consent and discontinue participation at any time without any consequence. To ensure confidentiality, we excluded personal identifiers, including names, addresses, phone numbers, and facility names, from all written materials. We restricted access to raw data, audio recordings, and transcripts to the research team only, protecting them through password-secured files and secure storage systems. We also presented findings in aggregated and summary form to ensure that individual participants could not be identified. All youth influencers included in the study were aged 18 years or older and were therefore treated as adults, with standard informed consent procedures. Young mothers aged 15–17 years were treated as emancipated minors and consented following the same procedure used for adult participants.

## Results

One hundred and ninety-eight participants were enrolled in the study, of whom 133 were FGD participants (Table 2).

The results are presented across six main themes: perceptions of immunization, sources of information on childhood vaccination, personal experiences of the consequences of non-immunization, barriers to immunization uptake, enablers of immunization uptake, and proposed intervention strategies.

## Perceptions of immunization

Participants reported that community awareness and attitudes toward childhood vaccination have generally improved over time, despite some individuals still holding on to past misconceptions. Most participants viewed vaccines as essential for preventing diseases, and many emphasized their role in supporting growth, development, and reducing morbidity and mortality. They highlighted the benefits of vaccination by comparing the higher prevalence of childhood illnesses in the past with the current lower rates. A 34-year-old mother of a fully immunized child from East Bale, Oromia region, described it as follows:

"*Many diseases affected our health, including our children's, in previous years due to lack of child vaccination, and currently the availability of vaccines prevented many diseases which made us happy. There was a negative attitude*

**Table 2. Number and characteristics of study participants.**

| Method | Type of participant | Number of participants | |
|---|---|---|---|
| | | Number of FGD, IDI, KII | Number of participants |
| Focus group discussions | Mothers | 8 | 18 FGDs 133 participants |
| | Men | 8 | |
| | Young mothers (15 – 17 years) | 2 | |
| In-depth interviews | Mothers of zero-dose children | 6 | 23 |
| | Mothers of under-vaccinated children | 5 | |
| | Mothers of fully vaccinated children | 6 | |
| | Mothers whose children are currently up-to-date but have not yet completed their immunization schedule | 6 | |
| Key informant interviews | Community leaders - kebele level | 7 | 42 |
| | Politicians – woreda/district level | 7 | |
| | Women leaders/ Youth influencers | 8 | |
| | Religious leaders/ Traditional healers | 8 | |
| | Community-based organization staff and represen-tatives of IDPs and people with disabilities | 5 | |
| | Health workers working on immunization | 7 | |

*toward vaccination previously and this belief and misconception was more common among our elders. Concerning the importance of vaccine it prevents the disease called 'dhukkuba qillensaa' (disease of air), 'lasheessa' (paralysis)] and diarrheal disease."*

## Sources of information on childhood vaccination

Community members from urban areas, along with key informants directly involved in immunization, were familiar with the names of some vaccines, although many could not specify which vaccine protects against which disease. Regarding information sources, urban participants and healthcare workers mentioned that formal education, training, the Ministry of Health website, television, radio, and social media are their trusted and commonly used sources of information on immunisation. In contrast, rural community members, as well as religious and community leaders, reported that healthcare workers—specifically Health Extension Workers (HEWs), the Health Development Army (HDA), and family members—were their most reliable sources of immunization information.

*"Our primary source of information regarding vaccination campaigns is the health professionals assigned to our local health post. They inform us when there is a vaccination campaign taking place."* (41-year-old, male, FGD participant, Fik district, Somali Region)

## Personal experiences of the consequences of non-immunization

Study participants shared various lived experiences regarding the consequences of non-immunization, either from their own families or observed within their communities. They mentioned frequent disease outbreaks, the death of family members, illness, and disability as some of the common outcomes of not vaccinating. A 25-year-old woman in an FGD from Dehana district in Amhara Region stated that:

*"I have personally experienced the consequences of not being vaccinated for measles. While all four of my brothers received the vaccination, I did not. During a measles outbreak in our village, I fell ill along with other children, and it*

 

*took several months for us to recover. However, my brothers remained unaffected by the disease and did not experience other illnesses. Now, I ensure that my children receive vaccinations right from birth to protect them from preventable diseases."*

Similarly, a 36-year-old key informant from Wadila district, Amhara Region, shared her experience, stating:

*"I share my experience of my oldest child who is suffering from paralysis. She is a 20-year-old beautiful young woman who cannot stand or walk. I accompanied her when visiting different health institutions including Addis Ababa Hospital. However, she didn't get [a] cure, and she was diagnosed with poliomyelitis because of not being vaccinated. Currently, she lives with a disability, and I am teaching the community members to vaccinate their children by showing the bad outcome my child had due to lack of vaccination"*

Another key informant shared a similar personal experience:
*"Two years ago, there was measles outbreak in our village. During that [outbreak], I lost my two children who were not vaccinated. Similarly, there were many deaths in the community due to the outbreak …. children who were vaccinated at that time faced less problems and most of them were alive."* (45-year-old, man, community leader, Shilabo district, Somali Region)

### Barriers to immunization uptake

We categorized barriers to immunization uptake into demand-side and supply-side challenges. Demand-side barriers pertain to issues faced by service users, while supply-side barriers result from the health system's inability to provide the required standard of care. We further classified these barriers into common issues across all study sites and inter-regional and region-specific challenges.

**Universal demand-side barriers.** We found that lack of awareness of immunization, fear of vaccine side effects, and mothers being busy with household chores were the most frequently cited barriers across the three regions.

*"One of the reasons why children may not be vaccinated is due to a lack of knowledge and awareness. Insufficient understanding of the importance of vaccines,[understanding] their benefits, and [understanding] the potential risks associated with not vaccinating can contribute to parents or caregivers choosing not to vaccinate their children. A lack of knowledge among mothers regarding the importance of childhood vaccination can lead them to decide not to vaccinate their children."* (59-year-old, man, religious leader, Fik district, Somali Region)

Another participant, a 21-year-old mother of an under-vaccinated child in Dehana district, Amhara, shared her views:
*"Initially, I vaccinated my child at 45 days old, but after the first dose, I [didn't follow the recommended] schedule because of the mild side effects like fever and irritation. Then, after three months following the regular schedule, I [got] the second dose [for my child]. Ultimately, I decided to stop vaccination entirely due to fear of side effects. I believe that my child became ill due to the side effects of vaccination. I'm hesitant to continue vaccination because of this fear."*
Mothers reported that being busy with daily household chores is a primary barrier in all study sites.

*"Since mothers are busy with their daily activities, they may [fail to] follow their child's vaccination schedule."* (52 year-old, man, community leader, Fik district, Somali Region)

**Inter-regional demand-side barriers.** A common inter-regional issue was the tendency to forget vaccination appointments, reported in both the Amhara and Oromia regions. A mother of an under-vaccinated child reflected on why she discontinued her child's vaccinations:

*"I forgot the schedule, (I have) no other problem (with vaccination)."* (29-year-old, mother of under-vaccinated child, Guradamole district, Oromia Region)

Community mistrust in the skills of vaccinators was also identified as a barrier in both Amhara and Somali regions.

*"Lack of [competency in skills] among healthcare professionals in administering injections is our concern."* (35 year-old, woman, FGD, Dehana district, Amhara Region)

Frequent changes in residential location were another common barrier reported by participants from the Oromia and Somali regions.

*"We have a community that moves from place to place. The houses are very far from the immunization site. Some urane [meaning temporary shelters that the pastoralist community constructs for temporary residency] may require a six-hour walk on foot to reach. So, such households may drop from the immunization schedule and expect to get immunization for the children after they return to the area."* (36-year-old, mother of fully-vaccinated child, Seweyna district, Oromia Region)

**Region-specific demand-side barriers.** In Amhara, common barriers to vaccination included the belief that a single vaccine dose is sufficient, reluctance to vaccinate until after the child's baptism, hesitancy to vaccinate unless therapeutic food supplements are provided, and fear of the 'evil eye.' The evil eye is a culturally held belief that a child can fall ill or even die if seen by someone outside the household, as certain gazes are thought to bring misfortune or sickness. To protect their children, caregivers often cover them with clothes or other items to prevent exposure to such harmful gazes

*"A prevalent cultural belief, such as the fear of the evil eye affecting their child in large gatherings, leads some caregivers to [consider] a single vaccine dose sufficient for a young child."* (41-year-old, mother of under-vaccinated child, Dehana district, Amhara Region)

Another participant also stated her challenge as a religious one, which also results in a longer delay to vaccination for girls compared to boys:

*"My child is not vaccinated because, as a culture in our community, vaccinating children before baptism is a religious misconduct (i.e., males are baptised 40 days and females are baptised 80 days after birth). So, I follow these trends and religious beliefs for all my children."* 28-year-old, mother of zero-dose child, Dehana district, Amhara Region)

In the Oromia region, a common barrier was the belief that vaccines exacerbate child illness.

*"My reason for interrupting vaccinating my child was the fear of the effects of the vaccine on the disease/illness the child currently has. My child has a disease called 'dhibee qilleensaa' (disease caused by air) which we believe can be exacerbated by vaccine injection."* (33-year-old, mother of under-vaccinated child, Seweyna district, Oromia Region)

Another frequently mentioned barrier in Oromia was the practice of not leaving home until the child reaches six months of age.

*"I think those not vaccinating their children [have] poor knowledge about the vaccination. There are beliefs in the community to not leave home for around six months after birth."* (36 year-old, woman, FGD, Seweyna district, Oromia Region)

In the Somali region, additional barriers reported included perceptions that vaccines are not important, beliefs that vaccines are harmful, and the view that some parents are negligent regarding their children's well-being.

*"Some mothers [reject] us and say vaccination is poison and don't kill our children and they are mothers of under-vaccinated or zero dose children. Some have a misunderstanding about immunization."* (25-year-old, female, health care worker, Shilabo district, Somali Region)

**Universal supply-side barriers.** Supply-side barriers across all study sites included issues such as distance and geographical barriers to health facilities in hard-to-reach areas, long waiting times, insufficient materials and equipment for vaccination, including vaccines, lack of compassionate and respectful care, a shortage of HEWs, as well as conflict, insecurity, and limited involvement of non-governmental organizations in vaccination locally.

In particular, long travel distances and waiting times pose a significant challenge for mothers with household chores and responsibilities competing for their time.

*"In the rural remote areas, the health extension worker appoints the mothers at a specific vaccination centre, but the health extension worker [travels a long time to reach the centre]. When the health extension worker reaches the area, many mothers would have left because of waiting a long time and getting disappointed. So, the difficulty of getting transport [for health workers] affects the vaccination services. In addition, even if they get a vehicle, they are unable to pay for the transport cost and decide to [cancel] the vaccination [service]."* (30-year-old, female, health care worker from Wadila district, Amhara Region)

In addition, the unavailability of vaccines at the health facility was mentioned as a barrier by mothers in FGDs.

*"I have visited this health post twice, but unfortunately, there has been a shortage of vaccines. The number of children requiring vaccination was more than the available supply of vaccines."* (17-year-old, young mother, Dehana district, Amhara Region)

Poor service delivery, including a lack of compassionate and respectful care, was a frequently cited challenge across all study sites.

*"There are delays in service provision, behavioural problems by the health professionals, not treating the [service users] equally, not respecting older people or the rural [residents], [health workers] who don't dress well or appear in good hygienic condition"* (43-year-old, man, religious leader, Wadila district, Amhara Region)

Participants across all study sites also highlighted the impact of frequent conflict and insecurity on child vaccination and other maternal and child health services. For instance, one FGD participant remarked:

*"The other factor that makes immunization service weak is war. It has been 3 to 4 years since Gashena has been a centre of conflict. Due to this reason, a pregnant woman in the rural area lacks pregnancy [care] services, gives birth at home and is not able to get her child vaccinated. So, I guess the war creates a big barrier for vaccination in our community - it is not a guess, it is a fact."* (39-year-old, man, Wadila district, Amhara Region)

**Inter-regional supply-side barriers.** In terms of regional commonalities, the most frequently mentioned challenge in both the Amhara and Oromia regions was the overreporting of vaccination coverage, which creates a perception of progress and limits efforts to address under-immunization.

*"There are issues of under-planning and overreporting of the number of vaccinated children in this particular district"* (32-year-old, female, government staff, Dehana district, Amhara Region)

Participants from Amhara and Somali regions reported challenges such as carelessness among HEWs, insufficient counselling on vaccine safety, and infrequent vaccination schedules, such as once-a-month appointments.

*"The timing isn't convenient as it's only offered one day per month, which is not suitable for everyone's schedule."* (41-year-old, mother of a child on immunization schedule, Dehana district, Amhara Region)

Participants in the Oromia and Somali regions reported that vaccinators often avoid opening new vials of BCG and measles vaccines when only a few children attend a session. This practice led to missed vaccination opportunities and was commonly identified as a problem.

*"To open measles vaccine at least there must be five children; if you open for one or two children, vaccines could be wasted. Dropout rate also increases as a result. Therefore, it is better if the vaccines are prepared in one dose for one child. Work must be done on measles and BCG vaccines."* (35-year-old, female, health care worker, Guradamole district, Oromia Region)

**Region-specific supply-side barriers.** In Amhara, non-functional HDA, and in Oromia, the residence of HEWs far from the communities they are meant to serve were frequently raised barriers.

*"[Inactive Health Development Army are] unable to effectively serve the community, primarily focused on seeking personal benefit packages. Additionally, most of them face challenges in reading and writing."*(40-year-old, man, FGD, Dehana district, Amhara Region)

*"Health extension workers usually live in town; health posts sometimes are closed due to this. I urge the government to follow up with health extension workers' presence at health posts and monitor actual service provision to the community. I think health posts should be opened [on] all days for better service."* (42-year-old, mother of zero dose child, Seweyna district, Oromia Region)

Also, in the Somali region, other challenges included the unavailability of vaccination services and a lack of awareness-raising efforts in certain areas.

*"In our community, children don't get vaccination due to unavailability of immunization services. Please provide us with the vaccines, and we all are ready to vaccinate our children."* (40-year-old, man, Shilabo district, Somali Region)

## Enablers of immunization uptake

Participants mentioned several factors that promote or encourage parents to vaccinate their children. We grouped these into demand-side and supply-side enablers.

**Demand-side enablers.** Frequently mentioned enablers included mothers sharing positive experiences with fully vaccinated children, their previous experiences with child vaccination, and witnessing the consequences of non-immunization.

*"The first time when I took my child for vaccination, mothers in the neighbourhood who had already immunized their children, emphasized the importance of vaccination and shared their positive experiences. Since then, I have taken*

*the responsibility of ensuring my children's well-being by bringing them to the health facility regularly for vaccinations."* (30-year-old, mother of child on immunization schedule, Fik district, Somali Region)

Another mother explained her motivator as:

*"The experience of witnessing the illness of an unvaccinated child in my neighbourhood strongly motivated me to ensure that my own child receives complete vaccination."* (29-year-old, mother of fully- immunized child, Dehana district, Amhara Region)

In Oromia, study participants frequently highlighted religious and community leaders' advice and active involvement as key enablers to immunization uptake.

*"There are a number of elders in the community regularly contacting, informing and encouraging health extension workers to work on addressing all community [groups] and vaccinating all eligible children."* (25-year-old, male, Seweyna district, Oromia Region)

In the Oromia region, men's involvement in vaccination efforts was recognized as a best practice. Typically, fathers play an active role by reminding mothers of vaccination dates, encouraging their participation, providing money for transportation, and, in some cases, taking the children to immunization sites themselves.

*"My husband, like other men, actively participates in the decision-making process regarding vaccination. He not only encourages but also reminds me of the importance of immunizing our child on schedule, emphasizing the timing of each vaccination."* (29-year-old, mother of child on immunization schedule, Seweyna district, Oromia region)

*"Basically, the father is the one who plays a huge role in putting pressure on the mother to take the child for vaccinations. Mothers are busy and have [other] workloads in their life, so they tend to forget about the vaccinations."* (42-year-old, community leader, Guradamole district, Oromia region)

**Supply-side enablers.** Many health workers reported that certain aspects of healthcare service delivery were seen as enablers, encouraging parents to vaccinate their children. In some areas, the commitment and efforts of health professionals and the HDA to disseminate vaccine-related information was identified as a key enabler.

*"As a health extension worker, my role to reach zero-dose children is to identify and search for such cases in the village……..we go 'Got' (smallest administrative unit below a village) to 'Got' with the help of health developmental army searching for zero-dose and under-vaccinated children."* (30-year-old, female, health care worker, Wadila district, Amhara Region)

Compared to previous years, the reduction in vaccine-preventable diseases within the community motivated parents to vaccinate their children as they witnessed the tangible benefits of vaccination.

*"Earlier, when there were no vaccination services, our children were facing different diseases like diarrhoea and vomiting, but now after we got [the immunization] service, there are no such diseases, even cough!"* (40-year-old, mother of child on immunization schedule, Guradamole district, Oromia Region)

Aligning vaccination dates with community preferences, such as on local holidays, was identified as a best practice in the Amhara region. This approach enabled parents, especially mothers, to vaccinate their children, as they were often busy with work on other days.

*"They plan [vaccination sessions] at a convenient time for the community, on the holidays, for example, making the vaccination days on 'St. Mikael' and 'St. Gabriel' (religious feast days celebrated in Orthodox Christian Ethiopian communities) and at the end of every month, so the community can go to vaccinate their children at a convenient time. These vaccination days are holidays; these days are not working days for the community. For example, the rural community might have to do farming during the working days."* (23-year-old, mother of child on immunization schedule, Wadila district, Amhara Region)

## Proposed intervention strategies

Study participants proposed interventions they deemed feasible for improving vaccination services in their respective areas. Across all three regions, participants highlighted the importance of raising awareness, ensuring the availability of sufficient materials and equipment, and maintaining a regular and adequate supply of vaccines to enhance immunization service utilization.

*"To sustain positive changes in awareness and attitude towards vaccination, ongoing education and outreach efforts are essential. This includes providing accurate information about vaccines, addressing concerns or misconceptions, promoting the benefits of vaccination, and engaging with community members through various channels such as schools, health clinics, religious institutions, and community events."* (22-year-old, female, health care worker, Tuliguled district, Somali Region)

Participants reflected on the importance of an adequate supply of resources, including vaccines.

*"Vaccines should be appropriately stocked in health facilities to avoid shortages and ensure that every child receives all required vaccines on time. Parents should not have to go far to get immunizations for their children."* (35-year-old, mother, Fik district, Somali Region)

Participants frequently suggested strengthening existing community structures, enhancing collaboration between the health system and the community, and engaging with community and religious leaders as key interventions to reduce zero-dose and under-immunized children.

*"The government needs to strengthen the women development army, the male development army, the youth association, and other community organizations to create awareness in the community regarding childhood vaccination so that it can be possible to improve the vaccination service in the community."* (30-year-old, female, health care worker, Wadila district, Amhara Region)

The importance of engaging community and religious leaders is also explained by this participant:

*"The desired immunization-related improvement can be brought about if we have discussions with influential tribal women [HaadhaSinke], influential tribal leaders [Abboti Gadaa]" and other influential people who can influence society to improve the uptake of immunization service utilization."* (32-year-old, female, government staff, Seweyna district, Oromia Region)

Expanding outreach vaccination services to remote areas, where most zero-dose children reside, was another widely suggested intervention across all study settings.

*"I recommend that the concerned office provide vaccination for [geographically remote] communities by regularly being present in the community. I suggest construction of extra health posts for [geographically remote] communities"* (30-year-old, mother of fully-immunized child, Seweyna district, Oromia Region)

Across all study sites, participants emphasized the importance of deploying sufficient, well-trained healthcare personnel capable of providing compassionate and respectful care as a crucial intervention to increase service uptake.

*"Creating a healthcare institution that is friendly and welcoming to caregivers. This involves ensuring a supportive and compassionate environment, where caregivers feel comfortable seeking vaccination services and can easily access the information they need."* (17-year-old, young mother, Dehana district, Amhara Region)

In the Amhara region, linking the continuum of maternity care with child vaccination was a frequently suggested intervention.

*"Starting from the antenatal care stage, it is crucial to register pregnant women and provide continuous support to ensure they deliver their babies at healthcare facilities. Timely commencement of immunization should also be emphasized."* (32 year-old, female, government staff, Dehana district, Amhara Region)

In the Somali region, establishing a reminder system was the most commonly recommended intervention.

*"I want to suggest establishing a system for tracking vaccination status and sending reminders to caregivers when their children are due for vaccinations. This could help ensure that children stay up to date on their immunizations."* (35 year-old, woman, FGD, Harshin district, Somali Region)

## Discussion

This formative study investigated the immunization experiences of diverse community groups in three regions in Ethiopia, focusing on the key factors that influence immunization uptake. We identified both universal and context-specific barriers and enablers to vaccination. While participants generally acknowledged the importance of vaccination, we discerned three primary categories affecting immunization uptake. First, knowledge and perceptions of vaccination and deeply rooted cultural and social norms emerged as critical elements across all regions. Second, the accessibility and convenience of immunization services to caregivers are fundamental to promoting vaccine uptake. Finally, personal and community experiences with vaccination and VPDs significantly impact attitudes and behaviours towards immunization.

We also found evidence that these three categories are inextricably connected and mutually affect each other. For instance, knowledge and perceptions of vaccination were often shaped by personal and community experiences with immunization and vaccine-preventable diseases, which in turn influence how caregivers interpret and respond to socio-cultural norms. Similarly, the accessibility and convenience of immunization services seemed to affect perceptions and vaccination intentions; even well-informed caregivers may delay or forgo vaccination if services are difficult to access [37]. Furthermore, positive service experiences can build trust, improve perceptions, and gradually shift community norms in favour of vaccination [38]. Together, these interacting factors highlight that immunization uptake is not determined by single or parallel domains, but by the continuous interplay between cognitive, social, and health system influences. These

insights are essential for triangulating evidence to enhance vaccination coverage and reduce the prevalence of zero-dose children in the study population.

## The influence of knowledge, perceptions and socio-cultural norms on vaccination uptake

We identified an intricate relationship between knowledge, misconceptions, and cultural norms affecting vaccination uptake among children in Ethiopia. We conceptualise knowledge as encompassing both accurate understanding and gaps in information, with misconceptions arising primarily from incomplete or incorrect knowledge, often reinforced by informal or unreliable sources [39]. In this context, while many caregivers were aware of the general benefits of vaccination, specific misunderstandings still persist. Most study participants emphasised the vital role of vaccines in preventing diseases in children. However, misconceptions about vaccines, such as one dose of multi-dose antigens being sufficient, persist among most community members. On the contrary, in the Somali Region, perceptions that vaccines are unimportant still persist. Our findings align with a recent nationwide study focusing on underserved populations in Ethiopia, which revealed that while knowledge and acceptance have improved among mothers over time, there remains a lack of comprehensive understanding, particularly regarding vaccine side effects and their management [40]. Other studies in Ethiopia have similarly reported misconceptions about vaccine timing, the appropriateness of vaccination during childhood illnesses, and the required number of doses [17,18,41]. Collectively, these findings suggest that despite an overall improvement in immunization knowledge, regional variations persist, indicating that awareness-raising interventions should be tailored to specific contexts.

Our findings also suggest that while participants reported a wide variety of trusted information sources about childhood vaccination, both health workers and health institutions, such as the Ministry of Health, were highly regarded in rural and urban areas, respectively. This finding also complements findings from two other studies in Ethiopia, one nationwide amongst underserved populations and one in Amhara Region, which also report that caregivers receive and trust vaccination information from health workers [40,42]. These results highlight that regular and meaningful interactions between caregivers and health workers, particularly in rural areas, present significant opportunities to improve perceptions of vaccination, address common misconceptions, and enhance understanding and awareness of immunization within the study communities.

We also identified social and cultural norms that hinder timely vaccination uptake. Although these norms vary by region, such as the tendency to wait for cultural milestones in Amhara and Oromia or the fear of the evil eye in Amhara, they operate through a similar mechanism: reducing the child's contact with conventional forms of vaccination service delivery modes, whether fixed or outreach. Our findings are consistent with other studies in Ethiopia reporting specific cultural norms and practices that reduce the contact of children under five years of age with modern health services, including the common use of traditional healers as the first point of care [43,44]. Given the late or minimal contact with health workers due to cultural norms, we argue that these further fuel misconceptions and negative perceptions of vaccination. Our findings show that involving religious and community leaders is perceived to have been successful in Oromia in shifting community attitudes toward vaccination and overcoming cultural hesitancies, and other studies have similarly emphasized the importance and potential of this approach [45,46]. However, our findings suggest that rather than only focusing on raising awareness for caregivers to abandon cultural norms to seek vaccination services through traditional fixed and outreach modes, innovations in vaccination service delivery that accommodate persisting cultural norms should be developed and piloted, focusing on increasing the contact of newborns and infants with the health system, and including strengthening the maternity care – childhood immunization continuum. These could, for example, include the provision of vaccination at home in parts of Amhara where infants do not leave the house until after baptism.

## The influence of service accessibility and convenience to caregivers on vaccination uptake

Our study identified that the accessibility and convenience of immunization services significantly influences vaccination uptake among caregivers in the study population. Barriers include infrequent vaccination schedules, unpredictable vaccine

availability, and long, difficult, or unsafe travel routes to the services. These are further intensified by the disproportionate responsibility placed on mothers, who are often solely accountable for ensuring childhood vaccination while simultaneously managing competing household demands, resulting in missed or delayed vaccination appointments. These barriers have been extensively documented as obstacles to immunization in Ethiopia [17,18,47–50]. Such challenges are particularly acute in rural, hard-to-reach, and conflict-affected communities, where health services are limited, making travel particularly unsafe. In our study, aligning vaccination schedules with local events or holidays has been reported as effective in the Amhara region, precisely because it provided convenient and accessible opportunities for caregivers to vaccinate their children.

We identified that service inaccessibility and inconvenience are exacerbated by health system capacity issues, especially a shortage of HEWs, logistics and infrastructure challenges in organizing timely and regular vaccination sessions, vaccine supply stockouts, and the reluctance of some vaccinators to open multi-dose vials for small numbers of children. These issues contribute to disruptions in already infrequent vaccination sessions, leading to longer waiting times and missed vaccination opportunities. Similar barriers have been reported in studies on immunization and community-based newborn care in Ethiopia [18,21,44,50,51].

Our findings suggest that strengthening immunization service delivery in the study population is crucial, specifically by addressing gaps in the availability and quality of human and material resources, and developing policies that balance vial opening with vaccine wastage reduction. However, in the short term, aligning vaccination schedules with local events, such as market days or holidays, along with increasing the frequency of vaccination sessions, implementing mobile vaccination service delivery (including house-to-house visits), and sending appointment reminders could significantly improve equitable vaccination coverage, particularly among zero-dose and under-immunized children.

## The influence of personal and community experiences with VPDs, vaccination and vaccination services

When sharing their experiences with under-immunization, participants frequently highlighted the severe consequences, such as disease outbreaks, death, and disability linked to non-immunization. These accounts align with those reported in a journalistic blog, where similar issues were highlighted in Ethiopia, including recurrent outbreaks, particularly of measles, as well as death and disability [52]. Furthermore, observing the positive health outcomes in fully vaccinated children within their communities served as a strong encouragement for others to follow suit, with peer influence —particularly among mothers—acting as a key enabler for vaccination uptake. The tangible benefits of vaccination, such as the reduction of VPDs, were recognized as important motivators, indicating that communities are aware of local VPD epidemiology. This is consistent with findings from other African countries, which report that communities can recognize disease outbreak events and other epidemiological changes in their communities [53–55]. These findings suggest that including real-life experiences from community members, including actively involving mothers of fully vaccinated children, and referencing improvements in local VPD burden during social mobilization efforts as cues to demand generation, may effectively improve immunization uptake in the community.

In contrast, negative experiences with immunization services, such as a lack of compassionate and respectful care from health workers, were a key barrier to immunization uptake. Reports of carelessness among HEWs and insufficient counselling on vaccine safety contributed to perceptions of inadequate service quality. In the immunization literature from Ethiopia, the association between caregiver satisfaction with immunization services and the quality of the interaction and communication with the vaccinators has been well-established [56–59]. Similarly, personal experiences of vaccination side-effects by caregivers discouraged return for future immunization. Caregiver hesitancy arising from personal or second-hand experiences of vaccination side effects has been previously reported as a reason for incomplete immunization in Ethiopia [17,40,50,60,61]. We argue that enhancing training for healthcare workers, coupled with supportive supervision, collecting monitoring data about patients' experience of care, addressing staff and material shortages, and ensuring a supportive work environment for vaccinators are crucial steps to ensuring compassionate, respectful care and

comprehensive vaccine counselling for caregivers. Such improvements could significantly increase caregivers' willingness to vaccinate their children and reduce the impact of concerns about vaccine side effects on hesitancy to complete future vaccinations.

## Study limitations

The study had several limitations. Most data collectors were male, which may have affected the openness of caregivers in discussing sensitive issues, such as decision-making power and male involvement in immunization decisions. Furthermore, the study was conducted during Ramadan, making it challenging to secure interviews in some of the study sites. The selection of participants at the village level by HEWs and community leaders may have introduced selection bias. While qualitative studies generally limit the generalizability of findings, we included a broad range of perspectives from caregivers, health workers, and community members, making the findings potentially relevant to similar settings in Ethiopia.

## Conclusion

Our research identified universal and region-specific barriers and enablers to immunization. The most salient factors influencing immunization uptake both negatively and positively were knowledge and perceptions of vaccination, deeply rooted cultural norms, accessibility and convenience of services, and personal and community experiences with VPDs, vaccination and vaccination services. Due to the strong interactions and synergies between the barriers, we recommend prioritising improvements in vaccination service delivery to improve immunization uptake, as this will likely result in synergistic effects addressing all three categories of barriers simultaneously. Specifically, we recommend diversifying vaccination delivery modes to increase service reach to children limited by cultural barriers, adapting the timing and frequency of vaccination sessions, and enhancing training, support, and resourcing for vaccinators to ensure compassionate and respectful care, as well as robust vaccine counselling. For social mobilization interventions, we recommend involving religious and community leaders and caregivers of fully vaccinated children and highlighting reductions in VPD burden.

## Supporting information

**S1 Text. Data collection tools: FGD, IDI and KII Guides.** This file contains the data collection instruments.
(DOCX)

**S2 Text. Consolidated criteria for reporting qualitative studies (COREQ): 32-item checklist.** This file contains the completed COREQ checklist for this study.
(DOCX)

**S1 Checklist. Inclusivity in global research checklist.** This checklist outlines the study's ethical, cultural, and scientific considerations specific to inclusivity in global research.
(DOCX)

## Acknowledgments

Our gratitude goes to the study participants for their time and valuable responses.

## Author contributions

**Conceptualization:** Catherine R. McGowan, Nada Abdelmagid, Yohannes Hailemichael.

**Data curation:** Amare Zewdie.

**Formal analysis:** Amare Zewdie, Minyahil Tadesse Boltena.

**Funding acquisition:** Alemseged Abdissa, Catherine R. McGowan, Nada Abdelmagid, Yohannes Hailemichael.

**Investigation:** Amare Zewdie, Minyahil Tadesse Boltena, Yohannes Hailemichael.

**Methodology:** Paula Valentine, Tahlil Ahmed, Catherine R. McGowan, Nada Abdelmagid, Yohannes Hailemichael.

**Supervision:** Amare Zewdie, Minyahil Tadesse Boltena, Mervat Alhaffar, Catherine R. McGowan, Nada Abdelmagid, Yohannes Hailemichael.

**Validation:** Minyahil Tadesse Boltena.

**Writing – original draft:** Amare Zewdie.

**Writing – review & editing:** Amare Zewdie, Minyahil Tadesse Boltena, Mengistu Ayenew, Tamrat Endebu, Melat Dereje, Alemseged Abdissa, Mirafe Solomon, Paula Valentine, Tahlil Ahmed, Andrew Clarke, Sostine Makunja, Mervat Alhaffar, Catherine R. McGowan, Nada Abdelmagid, Yohannes Hailemichael.

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
