## [Decision Letter · Decision Letter 0]

4 Dec 2025

PGPH-D-25-03042

Barriers to and enablers of childhood immunization uptake in Ethiopia’s Amhara, Oromia, and Somali Regions: A multi-perspective qualitative study

Dear Dr. Abdelmagid,

Thank you for submitting your manuscript to PLOS Global Public Health. After careful consideration, we feel that it has merit but does not fully meet PLOS Global Public Health’s publication criteria as it currently stands. Therefore, we invite you to submit a revised version of the manuscript that addresses the points raised during the review process.

Please note that we have only been able to secure a single reviewer to assess your manuscript. We are issuing a decision on your manuscript at this point to prevent further delays in the evaluation of your manuscript. Please be aware that the editor who handles your revised manuscript might find it necessary to invite additional reviewers to assess this work once the revised manuscript is submitted. However, we will aim to proceed on the basis of this single review if possible.

Could you please revise the manuscript to carefully address the concerns raised?

A rebuttal letter that responds to each point raised by the editor and reviewer(s). You should upload this letter as a separate file labeled ’Response to Reviewers’.A marked-up copy of your manuscript that highlights changes made to the original version. You should upload this as a separate file labeled ’Revised Manuscript with Track Changes’.An unmarked version of your revised paper without tracked changes. You should upload this as a separate file labeled ’Manuscript’.

We look forward to receiving your revised manuscript.

Kind regards,

Helen Howard

Staff Editor

Journal Requirements:

1. Please clarify all sources of funding (financial or material support) for your study. List the grants (with grant number) or organizations (with url) that supported your study, including funding received from your institution.

2. State the initials, alongside each funding source, of each author to receive each grant.

3. State what role the funders took in the study. If the funders had no role in your study, please state: “The funders had no role in study design, data collection and analysis, decision to publish, or preparation of the manuscript.”

4. If any authors received a salary from any of your funders, please state which authors and which funders.

3. In the online submission form, you indicated that Excerpts of the qualitative data supporting the findings of this study are available from the corresponding author upon reasonable request. However, full transcripts cannot be shared due to confidentiality agreements.

3. Uploaded as supplementary information.

Reviewers’ comments:

Reviewer’s Responses to Questions

**Comments to the Author**

1. Does this manuscript meet PLOS Global Public Health’s publication criteria? Is the manuscript technically sound, and do the data support the conclusions? The manuscript must describe methodologically and ethically rigorous research with conclusions that are appropriately drawn based on the data presented.

Reviewer #1: Yes

2. Has the statistical analysis been performed appropriately and rigorously?

Reviewer #1: N/A

3. Have the authors made all data underlying the findings in their manuscript fully available (please refer to the Data Availability Statement at the start of the manuscript PDF file)?

Reviewer #1: Yes

4. Is the manuscript presented in an intelligible fashion and written in standard English?

Reviewer #1: Yes

5. Review Comments to the Author

Reviewer #1: I have reviewed the manuscript and confirm that it meets the publication criteria of PLOS Global Health. The manuscript is presented in an intelligible fashion and written in standard English. I have attached my comments, which include certain statements that need to be revised to be clearer.

6. PLOS authors have the option to publish the peer review history of their article (what does this mean?). If published, this will include your full peer review and any attached files.

**Do you want your identity to be public for this peer review?** For information about this choice, including consent withdrawal, please see our Privacy Policy.

Reviewer #1: No

Figure Resubmissions:

---

## [Decision Letter · Decision Letter 1]

7 Apr 2026

PGPH-D-25-03042R1

Barriers to and enablers of childhood immunization uptake in Ethiopia’s Amhara, Oromia, and Somali Regions: A multi-perspective qualitative study

Dear Dr. Abdelmagid,

Thank you for submitting your manuscript to PLOS Global Public Health. After careful consideration, we feel that it has merit but does not fully meet PLOS Global Public Health’s publication criteria as it currently stands. Therefore, we invite you to submit a revised version of the manuscript that addresses the points raised during the review process.

One of the reviewers has raised a few comments. We kindly ask you to address these points and submit a revised version of your manuscript for further consideration.

A letter that responds to each point raised by the editor and reviewer(s). You should upload this letter as a separate file labeled ’Response to Reviewers’.A marked-up copy of your manuscript that highlights changes made to the original version. You should upload this as a separate file labeled ’Revised Manuscript with Track Changes’.An unmarked version of your revised paper without tracked changes. You should upload this as a separate file labeled ’Manuscript’.

We look forward to receiving your revised manuscript.

Kind regards,

Ifunanya Clara Agu

Academic Editor

**Journal Requirements:**

Reviewers’ comments:

Reviewer’s Responses to Questions

**Comments to the Author**

1. If the authors have adequately addressed your comments raised in a previous round of review and you feel that this manuscript is now acceptable for publication, you may indicate that here to bypass the “Comments to the Author” section, enter your conflict of interest statement in the “Confidential to Editor” section, and submit your "Accept" recommendation.

Reviewer #2: (No Response)

2. Does this manuscript meet PLOS Global Public Health’s publication criteria? Is the manuscript technically sound, and do the data support the conclusions? The manuscript must describe methodologically and ethically rigorous research with conclusions that are appropriately drawn based on the data presented.

Reviewer #2: Yes

3. Has the statistical analysis been performed appropriately and rigorously?

Reviewer #2: N/A

4. Have the authors made all data underlying the findings in their manuscript fully available (please refer to the Data Availability Statement at the start of the manuscript PDF file)?

Reviewer #2: Yes

5. Is the manuscript presented in an intelligible fashion and written in standard English?

Reviewer #2: Yes

6. Review Comments to the Author

**Reviewer #2:** In the ethical considerations section, I recommend clarifying the age range of the youth influencers included in the study.

I have two comments on the discussion section.

1) The authors write "We also found evidence that these three categories are inextricably connected and mutually affect each other." However, there is no additional discussion on how the categories are connected and mutually affect each other once they are described. I recommend adding in a short paragraph or a couple of sentences to demonstrate this point.

2) Regarding the first category on the influence of "knowledge, perceptions and socio-cultural norms on vaccination

uptake," I recommend clarifying differences between knowledge and misconceptions. For example, do people not know what the number of recommended doses of a vaccine are, or could you be more specific about what the misconception about this is (e.g., two are recommended but you only actually need one). If there is no difference, I recommend defining what you mean by knowledge (misconceptions and sources of information?) or changing the title of the section to reflect the focus on misconceptions and sources of information specifically.

7. PLOS authors have the option to publish the peer review history of their article (what does this mean?). If published, this will include your full peer review and any attached files.

**Do you want your identity to be public for this peer review?** For information about this choice, including consent withdrawal, please see our Privacy Policy.

Reviewer #2: No

**Figure Resubmissions:**

---

## [Editor Report · Decision Letter 2]

13 May 2026

Barriers to and enablers of childhood immunization uptake in Ethiopia’s Amhara, Oromia, and Somali Regions: A multi-perspective qualitative study

PGPH-D-25-03042R2

Dear Prof Abdelmagid,

We are pleased to inform you that your manuscript ’Barriers to and enablers of childhood immunization uptake in Ethiopia’s Amhara, Oromia, and Somali Regions: A multi-perspective qualitative study’ has been provisionally accepted for publication in PLOS Global Public Health.

If your institution or institutions have a press office, please notify them about your upcoming paper to help maximize its impact. If they’ll be preparing press materials, please inform our press team as soon as possible -- no later than 48 hours after receiving the formal acceptance. Your manuscript will remain under strict press embargo until 2 pm Eastern Time on the date of publication. For more information, please contact globalpubhealth@plos.org.

Best regards,

Ifunanya Clara Agu

Academic Editor